# Fabrication of a Novel Culture Dish Adapter with a Small Recess Structure for Flow Control in a Closed Environment

**Reiko Yasuda** [1,2] **, Shungo Adachi** [3] **, Atsuhito Okonogi** [2] **, Youhei Anzai** [2] **, Tadataka Kamiyama** [2] **, Keiji Katano** [2] **, Nobuhiko Hoshi** [1] **, Tohru Natsume** [3] **and Katsuo Mogi** [3,*]

[1]   Laboratory of Animal Molecular Morphology, Department of Animal Science,
      Graduate School of Agricultural Science, Kobe University, 1-1 Rokkodai, Nada-ku, Kobe 657-8501, Japan;
      r.yasuda@icomes.co.jp (R.Y.); nobhoshi@kobe-u.ac.jp (N.H.)
[2]   Icomes Lab Co., Ltd., 1-8-25 Kitaiioka, Morioka-shi, Iwate 020-0857, Japan; a.okonogi@icomes.co.jp (O.A.);
      y.anzai@icomes.co.jp (Y.A.); t.kamiyama@icomes.co.jp (T.K.); katano@icomes.co.jp (K.K.)
[3]   Molecular Profiling Research Centre for Drug Discovery, National Institute of Advanced Industrial Science
      and Technology (AIST), 2-4-7 Aomi, Koto-ku, Tokyo 135-0064, Japan; s.adachi@aist.go.jp (S.A.);
      t-natsume@aist.go.jp (T.N.)
[*]   Correspondence: mogi.k@aist.go.jp; Tel.: +81-3-3599-8251

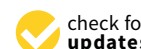

**Featured Application: The developed adapter makes it possible to spatially control flow in a commercially available culture dish in a manner previously only possible in microchannel devices. It will also be possible to create an arbitrary concentration gradient in a culture dish. Potential applications in a very wide range of fields include simultaneously reacting multiple single cells with various reagents in a culture dish and conducting drug discovery screening using very delicate concentration gradients.**

**Abstract:** Cell culture medium replacement is necessary to replenish nutrients and remove waste products, and perfusion and batch media exchange methods are available. The former can establish an environment similar to that in vivo, and microfluidic devices are frequently used. However, these methods are hampered by incompatibility with commercially available circular culture dishes and the difficulty in controlling liquid flow. Here, we fabricated a culture dish adapter using polydimethylsiloxane that has a small recess structure for flow control compatible with commercially available culture dishes. We designed U-shaped and I-shaped recess structure adapters and we examined the effects of groove structure on medium flow using simulation. We found that the U-shaped and I-shaped structures allowed a uniform and uneven flow of medium, respectively. We then applied these adaptors to 293T cell culture and examined the effects of recess structures on cell proliferation. As expected, cell proliferation was similar in each area of a dish in the U-shaped structure adapter, whereas in the early flow area in the I-shaped structure adapter, it was significantly higher. In summary, we succeeded in controlling liquid flow in culture dishes with the fabricated adapter, as well as in applying the modulation of culture medium flow to control cell culture.

**Keywords:** flow control; culture dish adapter; small recess structure; closed environment; perfusion culture

## 1. Introduction

The basic unit of an organism is the cell and an individual is formed by cells taking a more complicated structure to become tissues, organs, and organ systems. Cell culture is a technique for

growing these cells in an artificial environment outside the body. One of the major advantages of cell culture is that, in the environment that the cells grow in, physicochemical characteristics such as temperature, pH, and shear stress, as well as physiological components such as hormone and waste concentration, can be adjusted [1–3]. Moreover, it also makes it possible to analyze biological phenomena using a simpler experimental system compared to cellular analysis in vivo. Cell culture is a technology still in development: from culture dishes, a conventional technique, to perfusion culture in micro-channels for gene therapy and regenerative medicine and application in single-cell analysis and drug discovery screening [4–6].

Various conditions are required for culturing cells, but in terms of culture vessels, commercially available culture dishes are the most commonly used. In fact, when culturing cells, it is necessary to adjust the external environmental conditions, such as pH, $CO_2$ concentration, and temperature of the culture medium, for cells to grow in an environment most suitable for them, and these conditions will vary depending on cell type and experimental system. It is necessary to select the cell culture vessel, such as culture dishes or flasks, according to the desired purpose [7,8]. However, what is widely used in laboratories are the commercially available culture dishes sold by manufacturers [9]. Each laboratory has a detailed know-how of each product, such as adhesion of cells to culture dishes, and even among commercially available culture dishes, each researcher has his or her own preference.

Advances in microfluidic device technology have led to progress in research on perfusion culture [10], and it is becoming clear that liquid flow has a large influence on cells. Presently, owing to the progress in microfabrication technology, it is possible to create microscale fluidic devices that are very complex and have a high difficulty of fabrication [11], and that allow reproducible cell culture experiments within microchannels [12–14]. In fact, studies using microfluidic devices showed that stem cells and iPS cells/ES cells are influenced by shear stress, and this promotes differentiation [15–17]. Therefore, the influence of culture medium flow during cell culture is very large. If this flow can be controlled, researchers can modulate the influence of flow on cells, particularly shear stress. However, in culture experiments using microfluidic devices, a skilled technician is required to deliver liquid to the microchannel, and contamination from connection points in the complex structure of the device is often a problem [18]. Thus, unsolved problems remain in culture experiments using microfluidic devices for practical applications owing to the complexity of the equipment.

An efficient perfusion culture equipment using culture dishes remains to be developed [19], and it is difficult to control the flow in currently used culture dishes. In perfusion culture experiments using commercially available culture dishes, problems such as flow rate accuracy of the tube pump and meniscus in the culture dish remain to be solved, and a method for controlling the flow in the culture dish has not been established. In typical tube pumps, errors in the inner diameter of the tube affect the amount of liquid delivered [20], and when injection and discharge are controlled using a tube pump, it is difficult to keep the injection and discharge amounts constant, leading to stagnant liquid or leakage in the culture dish. In addition, because the culture dish itself is an open type chamber, a meniscus occurs between the dish and the medium, which results in a faster flow of culture medium along the edge of the dish owing to the property that liquid flows more easily toward the higher liquid bulk volume, and thus it is very difficult to control flow in a culture dish [21].

If these problems are resolved, it becomes possible to use, in combination with a commercially available culture dish, a device manufactured using laboratory equipment that arbitrarily controls the flow in a culture dish, which is difficult to realize with a commercialized device [19]. In addition, it will be possible to assemble and manufacture a simple handling device and evaluate the influence of fine flow in a macro culture environment that is difficult with microfluidic devices [9–11,22]. In other words, it becomes possible to spatially control flow in a commercially available culture dish in a manner previously only possible in microchannel devices. It will also be possible to create an arbitrary concentration gradient in a culture dish (Figure 1). Potential applications in a very wide range of fields include simultaneously reacting multiple single cells with various reagents in a culture dish and conducting drug discovery screening using very delicate concentration gradients.

In this study, we fabricated a polydimethylsiloxane (PDMS) culture dish adapter (CD-adapter) that enables fluid control with a small recess structure, and also investigated whether flow control affects cultured cells.

## 2. Fabrication of the PDMS CD-Adapter

We fabricated an adapter with a small recess structure to control the flow of liquid in a culture dish in a closed environment. To fabricate a PDMS adapter that can be attached to a commercially available culture dish without gaps, a method using a culture dish equipped with a negative structure of a small recess structure as a mold was adopted. A U-shaped small recess structure (U-shaped structure) and an I-shaped small recess structure (I-shaped structure) as shown in Figure 1 were provided in a CD-adapter corresponding to a 35-mm culture dish. A semicircular arc with outer and inner diameters of 32 mm and 30 mm, respectively, and a rectangle with a width and length of 2 mm and 34 mm, respectively, were cut out from a 500-μm thick polyethylene sheet (HRHG711303, KYOWA). The circular arc was pasted on the bottom of a 35-mm culture dish with a 2-mm gap from the edge, and the rectangle was stuck on a line bisecting the culture dish (Figure 2A). The outer periphery of the 35-mm culture dish with a mold structure was filled with epoxy resin (STYCAST 126J PTA, Henkel) for the bearing surface of the adapter. Therefore, as shown in Figure 2A, a 60-mm culture dish was used as a tray, being attached centered under the 35-mm culture dish. Epoxy resin was poured into the gap between both culture dishes, which were left for 9 h to stand at 25 °C to cure the epoxy resin (Figure 2B). After checking that the epoxy resin had cured, 2 mL of PDMS (CAT-106F, Shin-Etsu Chemical) was injected into the 35-mm culture dish and left to stand at 25 °C for 6 h to cure the PDMS (Figure 2C). The reason for setting the volume of PDMS to 2 mL is to make the thickness of the bottom membrane of the CD-adapter 2 mm. Because the bottom area of the 35-mm culture dish is 9 cm$^2$, the volume of PDMS required to make the bottom membrane thickness 2 mm is 1.8 mL. In this study, we set it to 2 mL for easy preparation of the experiments. After confirming that the PDMS had cured, the 35-mm culture dish and the center of a cylinder (aluminum cylinder, Showa Denko, Tokyo, Japan) with a diameter of 25 mm was placed on the PDMS membrane (Figure 2D), and PDMS was poured in a 60-mm culture dish so as not to overflow the edge (Figure 2E). After leaving it at 25 °C for 9 h to cure the PDMS, the aluminum cylinder and culture dish were peeled off, and the molded PDMS taken out. Using a biopsy punch (BP-L20K, Kai, Tokyo, Japan), a 2-mm diameter hole was drilled at the intersection between the circular arc of the small recess structure and the Y-Y' part of the fabricated PDMS adapter, and at a linear small recess structure 2 mm away from the edge of the adapter, to serve as an inlet. Similar holes were made on each opposite side to form an outlet (Figure 2F,G). The completed sealed adapter was a cylindrical-shaped PDMS with a 2 mm wide, 500 μm deep recess structure on a 2 mm thick bottom membrane, and an inlet and outlet with an inner diameter of 2 mm (Figure 2G).

The fabricated CD-adapter had a very high transparency and was composed of PDMS only, and thus it has good oxygen permeability and a very suitable structure for cell culture. Furthermore, because the central membrane portion was a thin transparent membrane with a thickness of 2 mm, it was also possible to observe the culture dish with the adapter attached on it with a microscope (Figure 3A). The fabricated U-shaped and I-shaped structures are shown in Figure 3B,C, respectively.

As shown in Figure 3D, a transparent framework made of acrylic resin was used as a holding jig to prevent liquid leakage. It has a circular shape so that a constant pressure can be applied to the side of the culture dish for tight sealing. In addition, the center of the culture jig contains a hole with a diameter of 10 mm through which cells can be observed using a microscope. The holding jig has a height of 5 mm and an outer diameter of 27.0 mm. Furthermore, to adjust the amount of medium in the culture dish, a ring spacer was placed between the CD-adapter and the culture dish to raise the adapter. Because the CD-adapter was molded from the culture dish, when the CD-adapter is attached to it, the bottom surface of the CD-adapter touches the bottom of the culture dish. In this study, to set the amount of medium in the culture dish to 2 mL, the ring spacer was made of silicone with a thickness of 2 mm and the CD-adapter was lifted 2 mm from the bottom of the culture dish.

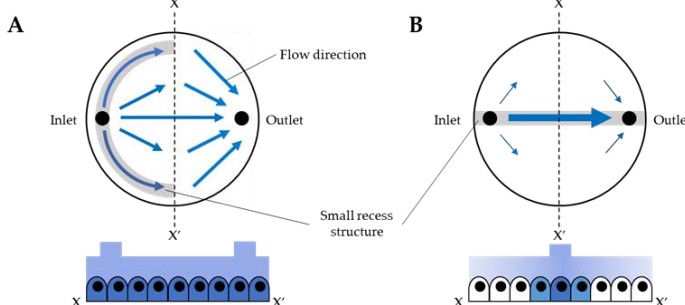

**Figure 1.** Flow control in the closed environment of a culture dish. By providing a structure that can control flow in a culture dish by adapter selection, it is possible to arbitrarily modulate the liquid flow in the closed environment without changing the culture medium, culture dish, or injection rate. (**A**) Delivered liquid flows uniformly from the inlet towards the outlet at an almost constant speed irrespective of the position in the culture dish. Cells in culture dishes can be stably cultured under the same conditions. (**B**) Delivered liquid flows linearly from the inlet to the outlet. A concentration gradient occurs in the culture dish along the flow of the delivered liquid, and, as a result, it is possible to modulate its influence on cells by adapter selection.

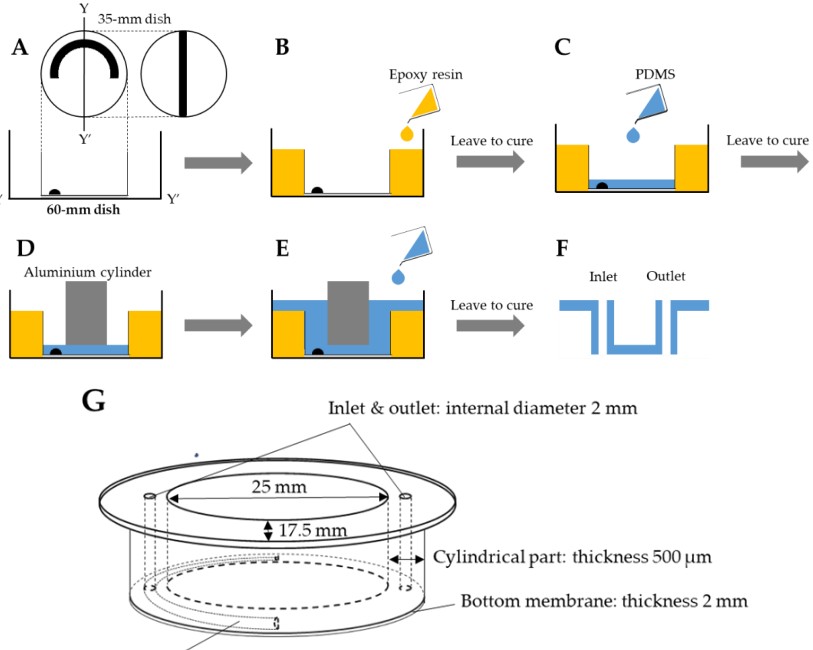

**Figure 2.** Manufacturing method of the culture dish adapter (CD-adapter). (**A**) Cut out a circular arc of 2.0 mm of width and 32 mm of outer diameter from a polyethylene sheet with a thickness of 500 μm and paste it on the bottom of a 35-mm culture dish from Nunc and 2.0 mm from the edge. Place the 35-mm culture dish to which the circular arc was attached in the center of a 60-mm culture dish using double-sided tape. (**B**) Pour epoxy resin into the 60-mm culture dish and leave at 25 °C for approximately 9 h to cure. (**C**) Pour 2 mL of polydimethylsiloxane (PDMS) into the 35-mm culture dish and let it stand at 25 °C for approximately 6 h to cure. (**D**) Place an aluminum cylinder with a diameter of 25 mm at the center of the 35-mm culture dish. (**E**) Pour more PDMS around the aluminum cylinder and leave at 25 °C for approximately 9 h to cure. (**F**) Remove the aluminum cylinder and culture dish from the PDMS and use a long-type biopsy punch with a diameter of 2 mm to make holes for injection and discharge. (**G**) CD-adapter made of PDMS compatible with a 35-mm culture dish. The adapter has an inlet and an outlet and a concave groove at the bottom for equalizing the flow in the culture dish. The depth of the groove is 500 μm, the diameter of the inlet and outlet is 2.0 mm, and the membrane thickness of the bottom surface of the adapter is 2.0 mm.

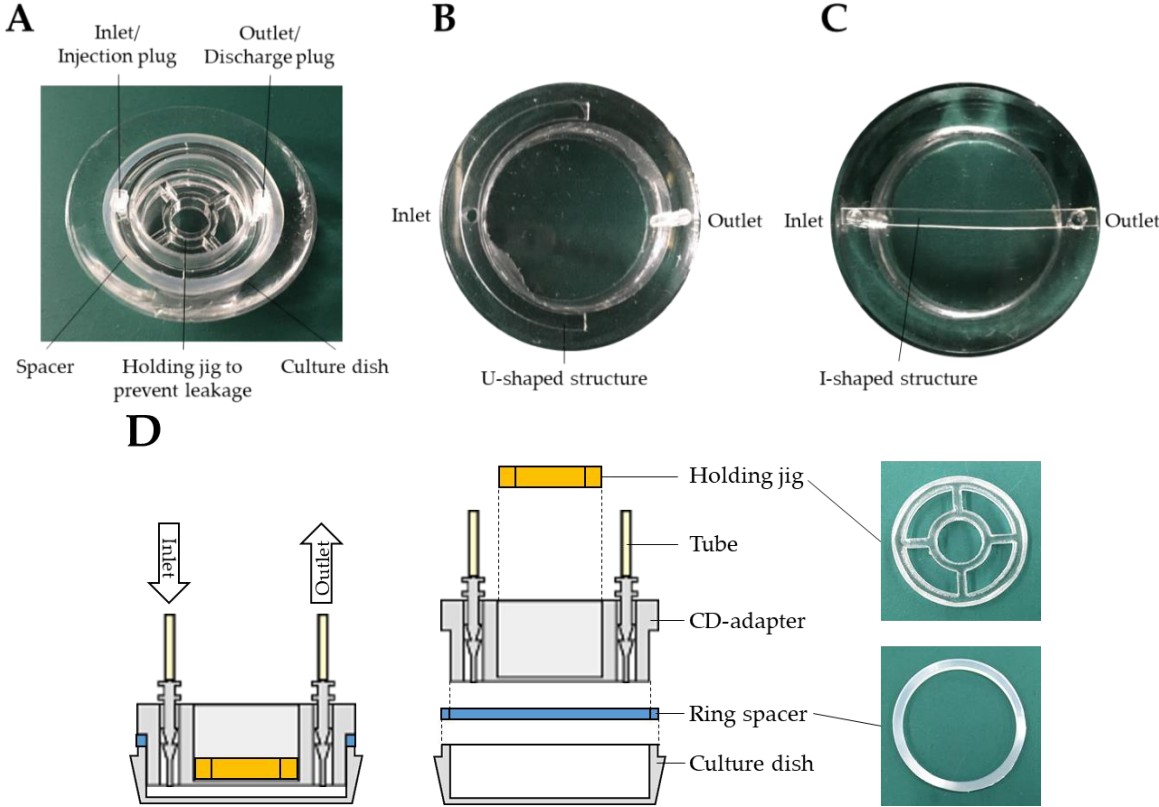

**Figure 3.** The fabricated CD-adapter with its small recess structure. (**A**) A CD-adapter compatible with 35-mm culture dishes. Because it is fabricated with polydimethylsiloxane (PDMS), it is highly transparent and enables cell observation by microscope. In addition, owing to good air permeability, it can supply enough oxygen to cells even in a sealed state. A plug was inserted into both inlet and outlet to connect to the tube, and the amount of medium in the culture dish can be adjusted by raising the adapter with a spacer. Furthermore, a holding jig to prevent liquid leakage was fitted in the center of the adapter, and pressure is applied from the inside toward the culture dish to increase adhesion between the adapter and the culture dish. (**B**) Small recess structure created on the bottom of the CD-adapter used in this work. A horseshoe-like small recess structure spreads from the inlet side to the equatorial line in the bottom. The width of the groove was 2.0 mm and its depth was 500 μm. (**C**) Small recess structure created on the bottom of the CD-adapter used in this work. A linear small recess structure extends from the inlet side to the outlet side. The width of the groove was 2.0 mm and its depth was 500 μm. (**D**) Procedure for setting the CD-adapter to the culture dish. A ring spacer of 2 mm thickness made of silicon was placed between the culture dish and the CD-adapter to raise the adapter and make cell culture space. A holding jig was fitted in the center of the adapter to prevent liquid leakage.

## 3. Experiment

### 3.1. Fluid Simulation of Flow Control with the CD-Adapter

Perfusion culture was performed using a CD-adapter with two types of small recess structure, and flow control was measured in each cell adhesion area. Before performing culture experiments, fluid simulation of the flow control of the small recess structure using commercial software of finite element method (COMSOL, KESCO) was performed. Figure 4A shows the flow velocity distribution in an adapter with no small recess structure when the flow rate was 1.4 μL/min, and Figure 4D shows the flow velocity distribution in the Z-Z′ cross section. Since there was a strong gradient in the flow rate in the culture dish unless a small recess structure was provided in the adapter (Figure 4D), we designed a U-shaped structure that can maintain an almost uniform flow in the culture dish. And to more

clearly demonstrate that flow control is possible, we also designed an I-shaped structure with a more pronounced flow bias compared to the U-shaped structure. Similarly, Figure 4B,C show flow velocity distributions in the U-shaped and I-shaped structures. Figure 4E,F show flow velocity distributions in the Z-Z' sections in Figure 4B,C. In Figure 4F, the flow of liquid increased markedly in the I-shaped structure compared to that shown in Figure 4E. From these results, we conclude that the flow velocity distribution in the culture dish can be controlled by the shape and position of the small recess structure.

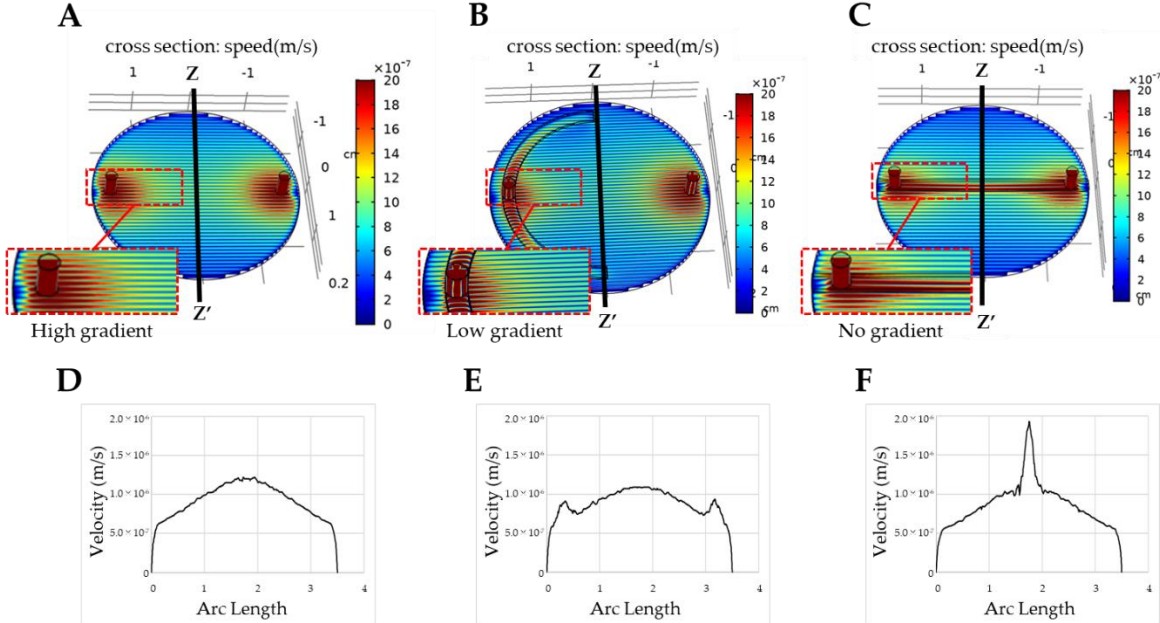

**Figure 4.** Simulation analysis of liquid flow in a perfusion culture dish with a CD-adapter. Liquid flow in the culture dish at 1.4 μL/min through the inlet was simulated and analyzed using COMSOL. (**A**), adapter without the small recess structure; (**B**), U-shaped structure adapter; (**C**), I-shaped structure adapter. Flow velocity on the Z-Z' line in (**A**–**C**) are plotted in (**D**–**F**), respectively. In (**F**), the liquid flow remarkably increased under the I-shaped groove compared to (**E**).

### 3.2. Cell Culture Test for Flow Controllability of the Small Recess Structure

Based on the results of the simulation, three adapters were prepared for each U-shaped and I-shaped structure. Cells from the 293 T cell line, which are kidney cells derived from a human fetus, were seeded in 35-mm culture dishes (IWAKI) at a seeding density of $1 \times 10^5$ cells/mL and cultured for 1 day in a 37 °C—5% $CO_2$ incubator. The culture medium was prepared by adding 10% fetal bovine serum (FBS) to Dulbecco's modified Eagle's medium (DMEM), and 2 mL was added to the culture dish.

The culture medium of cells cultured for 1 day was completely removed and replaced with serum-free DMEM supplemented with no FBS, the lid of the culture dish was removed, and a CD-adapter, which was autoclaved and dried, was attached to the dish. Next, the microtube pump system (Icomes Lab, Iwate, Japan), the reservoir, and the waste reservoir were connected to the clean bench as shown in Figure 5. DMEM supplemented with 10% FBS was perfused at a flow rate of 17 μL/min, which did not influence the shear stress, for 40 min by operating the pump system [14], and then the pump was stopped and the cells were cultured for 1 day at 37 °C—5% $CO_2$. The cells were then imaged at a magnification of 40× with a microscope (inverted microscope IX71, OLYMPUS, Tokyo, Japan) at points i, ii, and iii in Figure 6A. Images were processed using ImageJ, and the area of adhered cells was plotted (Figure 5).

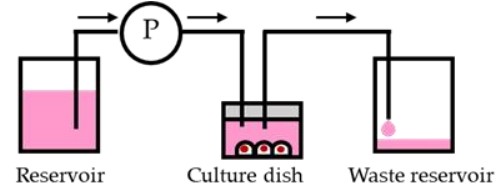

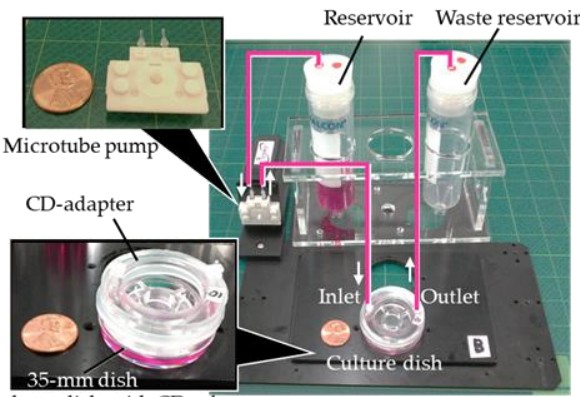

**Figure 5.** Schematic of the culturing system and photo of the experimental setup. A CD-adapter was attached to a 35-mm culture dish on which cells had been seeded, and connected to a micro tube pump system, reservoir, and waste reservoir. The size of the experimental setup is $200 \times 150 \times 150$ mm, which can be installed on a stage of the microscope and also can be installed in the incubator.

## 4. Results and Discussion

Culture experiments were carried out three times (N = 3), and the adhesion area of cells after 3 days was averaged for each observation point and plotted. Results in i and iii are normalized by value of cell adhesion area in ii in Figure 6A, and the error bars correspond to standard deviation. Adhesion areas of cells in i, ii, and iii was 0.92, 1.0, and 0.92, respectively, in the U-shaped structure adapter, and 0.65, 1.0, and 0.73, respectively, in the I-shaped structure adapter.

It is usually necessary to add serum to the culture medium of 293T cells, which otherwise do not grow normally and die. Therefore, cells over which serum-containing medium flowed should have proliferated more than those at sites with no flow. As can be seen from the simulation results in Figure 4B, culture medium flowed evenly through the culture dish from the inlet to the outlet in the U-shaped structure. Therefore, it can be assumed that, by delivering culture medium with serum, the cells will grow uniformly throughout the culture dish. Indeed, as a result of delivering the serum-containing culture medium, the cells showed the same level of proliferation in all areas of the culture dish (Figure 6B). In contrast, the simulation results in Figure 4C show that the flow velocity of the culture fluid became faster on the straight line under the small recess structure of the I-shaped structure. Accordingly, the cell culture results show that the cell proliferation rate was higher in ii, immediately below the small recess structure, compared to other areas (Figure 6B). The culture and simulation results are consistent, and we conclude that flow control can be exerted with cell culture evaluation using the small recess structure provided in the CD-adapter.

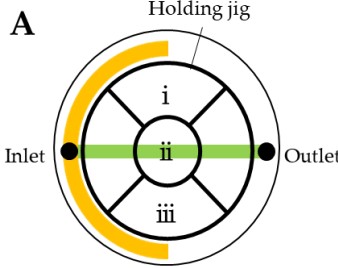

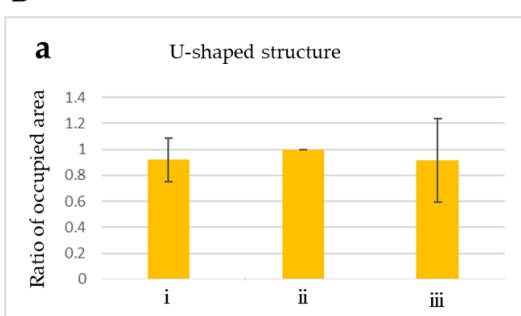
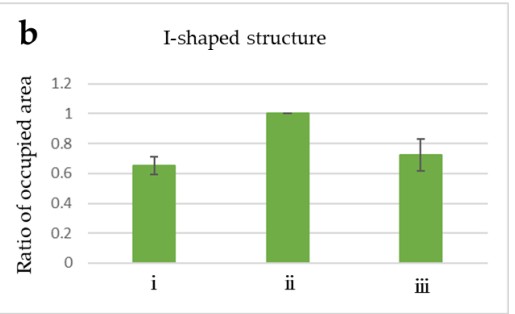

**Figure 6.** Culture of 293T cells using the CD-adapter. (**A**) Cells were observed at points i, ii, and iii on the vertical line intersecting the center of the straight line that connects the inlet and the outlet. DMEM culture medium supplemented with 10% fetal bovine serum was perfused at a flow rate of 17 μL/min through the inlet for 40 min, and the cells were then cultured for 1 day. (**B**) Cultured cells were imaged at all points, and the area occupied by cells was quantified using ImageJ software. The area occupied at ii was set as 1, and then the relative areas at i and iii were obtained. With the adapter with the I-shaped structure (**b**), the rate of cell proliferation under the small recess structure was significantly higher than that in other places, and no difference in proliferation rates was observed with the adapter with the U-shaped structure (**a**) at points i–iii, and it was possible to culture cells uniformly.

## 5. Conclusions

In this study, we developed a CD-adapter with recess structure to exert flow control in a closed environment using commercially available culture dishes. We successfully fabricated the CD-adapter using inexpensive PDMS by an easy process without any special equipment for microfabrication. The flow controllability of the recess structure was estimated using the finite element method and demonstrated by cell culturing in the closed chamber to confirm that flow control has an influence on cultured cells. Although precise control of microscale spatial resolution was not obtained, liquid flow was modulated using the developed CD-adapter with a small recess structure, and cell culture was controlled by modulating the flow of culture medium.

A more delicate control of the flow may be obtained by further elaborating the shape of the small recess structure of the adapter, and adapters for various culture dishes can be fabricated cheaply and freely. In the future, we will evaluate the flow in different shapes of the small recess structure in various adapters and their influence on cells. As this research progresses, researchers and research institutes conducting cell culture experiments will be able to perform more sophisticated experiments without greatly changing their current experimental setup, and it may become possible to conduct complex research that, until now, could only be performed using microfluidic devices in a macro environment.

**Author Contributions:** Conceptualization, R.Y., K.M., and S.A.; formal analysis, R.Y. and K.M.; funding acquisition, O.A., S.A., K.K., and T.N.; investigation, R.Y.; methodology, R.Y. and K.M.; project administration, R.Y.; resources, Y.A. and T.K.; supervision, N.H. and T.N.; validation, R.Y.; visualization, R.Y.; writing—original draft preparation, R.Y.; writing—review and editing, M.K., S.A., N.H., and T.N.

**Funding:** Part of this perfusion system was developed with financial support from the Grant-in-Aid for Sample creation support project No. 408005 of the New Energy and Industrial Technology Development Organization (NEDO) of Japan and the Adaptable and Seamless Technology Transfer Program through Target-driven R&D

(A-STEP) No. AS2815006U from the Japan Science and Technology Agency (JST). This work was partially supported by Grant-in-Aid for young scientists (B) (No. 17K17715) and Leading Initiative for Excellent Young Researchers (LEADER) in FY 2016 of the Ministry of Education, Culture, Sports, Science and Technology, Japan.

**Acknowledgments:** The authors would like to thank H. Kotera and H. Shintaku (Institute of Physical and Chemical Research) for assistance with the numerical simulations.

**Conflicts of Interest:** The authors declare no conflict of interest.

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
