# Peer review of "Fabrication of a Novel Culture Dish Adapter with a Small Recess Structure for Flow Control in a Closed Environment"

_applsci, doi:10.3390/app9020269_

Round 1
Reviewer 1 Report
In this paper, the authors fabricated a culture dish adapter using PDMS that has a U-shaped or I-shaped recess microstructure for flow control. They examined the effects of groove structure on medium flow using both simulation and experiments. The authors succeeded in controlling liquid flow in culture dishes with the fabricated adapter, and examined the effects of recess microstructures on cell proliferation. This method is interesting; however, the following comments need to be addressed before recommending for publication:
1. In Figure 1 A, the blue arrows are confusing, do they represent the direction of flow?
2. Figure 3 and Figure 5 didn’t clearly show the structure of the adapter and the connection between the adapter and culture dish. It’s confusing how the fluid flow through the recess microstructure and reach the culture dish. The authors should show a more detailed and enlarged figure clarifying the structures of the adapter and the whole system.
3. In Figure 3, the authors claimed that “the amount of medium in the culture dish can be adjusted by raising the adapter with a spacer”, and “a holding jig to prevent liquid leakage was fitted in the center of the adapter, and pressure is applied from the inside toward the culture dish to increase adhesion between the adapter and the culture dish.” It is necessary for the authors to illustrate the details regarding the adjustment of the spacer and the holding jig.
4. The authors also should clarify the advantages of current device for flow control in comparison with commercialized ones or using microfluidic methods. Also the introduction part should include recently published methods medium exchange techniques (e.g. Electrophoresis, 2016, 37, 2147; Analytical Chemistry, 2017, 89, 9574; Analytical and Bioanalytical Chemistry, 2015, 407, 3437).
5. Can the authors explain theoretically why the recess microstructure has influence on the flow speed? Why the authors chose to study the U-shaped and I-shaped recess microstructure instead of other shapes?
5. During the perfusion process, will cells maintain steadily inside the culture dish? Is there any cell loss due to the fluid flow?
6. In Figure 4, authors conducted simulation analysis of liquid flow in a perfusion culture dish with a CD-adapter at 1.4 μL/min through the inlet, how the speed in the cross section change with the variation of the flow rate from inlet?
7. Authors should include more comparison experiments such as the comparison with or without the recess structure, comparison between perfusion cell culture and in static conditions.
8. There are some typos and grammar mistakes in this paper. The authors should read carefully and double check the language.
Author Response
Response to Reviewer 1 Comments
December 27th, 2018
From: Reiko Yasuda
Graduate School of Agricultural Science, Kobe University
1-1 Rokkodai, Nada-ku, Kobe, 657-8501, Japan
Tel: +81-78-803-5928
r.yasuda@icomes.co.jp
Dear Sir,
We thank you for careful reading our manuscript and for giving useful comments. Your valuable comments helped me very much to improve the quality as well as the readability of the paper. On the basis of your comments, I added new description. I hope the revision could meet the conditions to be considered as the publication. Please see attached word file for details of response to comment.

Reviewer 2 Report
R. Yasuda and coauthors present a perfusion system dedicated to commercially available 35-mm culture dishes. The polydimethylsiloxane part is placed inside the Petri dish so that the flow velocity of the cell medium is supplied either almost uniformly or through a gradient by a U-shaped or I-shaped recess structure, respectively. The efficiency of the perfusion system in the culture dish is demonstrated both by simulations of the fluid flow and by measurements of confluence in the Petri dish. The introduction of the manuscript highlights the influence of shear stress by the liquid flow onto the cell fate, which was actually observed in the difference of cell growth between the various areas of the culture dish.
Generally speaking, the manuscript is well written. The figures are informative. The description of the fabrication process is clear. In the reviewer’s opinion, the manuscript should be accepted to publication after providing some revisions:
- The term “recess microstructure" is used in the title and in many parts of the manuscript. As the recess structure is 2 mm large, several millimeters long and 500 µm thick, “microstructure” seems to be inaccurate. “millimeter-scale recess structure” or “small recess structure” may be more appropriate.
- The introduction should include a bibliography about cell culture systems developed for commercial culture dishes. For example:
* E. Kondo, K.-I. Wada, K. Hosokawa, M. Maeda. Microfluidic perfusion cell culture system confined in 35 mm culture dish for standard biological laboratories. Journal of Bioscience and Bioengineering, 2014, 118(3), 356-358. This article shows a gravity-driven medium perfusion system, a microfluidic filter and a cell collection and culture chamber made in a 35 mm culture dish.
* S. Y. Lee, S. Yang. A microfluidic-based lid device for conventional cell culture dishes to automatically control oxygen level. BioTechniques, 2018, 64(5), 231-234. This article presents a lid made by eight PDMS layers to be placed onto a culture dish to control the oxygen level dissolved in the cell medium.
- It is unclear in the manuscript whether the Culture Dish-adapter is in contact or not with the bottom of the culture dish. If so, how can the liquid flow under the Culture Dish-adapter and reach the cell culture?
The legend of the Figure 3 reads that "the amount of medium in the culture dish can be adjusted by raising the adapter with a spacer" (Page 4 Lines 149-150). Where is the position of the spacer: between the top membrane of the Culture Dish-adapter and the vertical wall of the 35-mm culture dish? As the position of the spacer is not clear enough in the Figure 3A, this information should be provided in the main text.
If a spacer is present, what is the thickness of the spacer? This is important information because the thickness of the spacer will determine the height of the chamber for the cell population.
- Page 4 Lines 150-152 (legend of the Figure 3), "a holding jig to prevent liquid leakage was fitted in the center of the adapter, and pressure is applied from the inside toward the culture dish to increase adhesion between the adapter and the culture dish"
+ Page 6 Lines 185-187, "To prevent liquid leakage, a holding jig was installed inside the cylindrical cavity to increase the degree of sealing between the adapter and the culture dish"
+ Page 6 Lines 195-197(legend of the Figure 5), "to prevent liquid leakage, a holding jig was installed between the adapter and the culture dish to ensure sealing between the adapter and the culture dish installed inside the cylindrical cavity":
Is the holding jig applying some pressure onto the bottom membrane of the PDMS adapter or onto the vertical wall of the Culture Dish-adapter? In other words, must the Culture Dish-adapter be sealed to the bottom or the vertical wall of the culture dish to prevent any leaking?
In case the pressure by the holding jig is exerted onto the bottom of the culture dish, i.e., if the CD-adapter and the bottom of the culture dish are sealed, how can the liquid flow go out of the recess structure?
- Page 3 Lines 100-102: please indicate that the 2 mL of PDMS poured into the 35-mm culture dish (Figure 2C) result in a 2-mm thick PDMS layer, so that the readers don’t have to calculate the PDMS thickness by themselves. This clarification will help the readers to understand that 1) the 500 µm thick recess structure is entirely covered by the PDMS layer, and 2) the poured PDMS is the origin of the 2 mm thick bottom membrane mentioned in the Figure 2G.
- Figure 1A: the blue arrows on the right part of the culture dish (i.e., on the right of the XX’ line) are badly oriented. These arrows should point to the outlet, rather coming from the outlet.
- Page 3 Lines 114-116 (legend of the Figure 1), "it is possible to modulate the liquid flow in the closed environment without changing the culture medium, culture dish, or injection rate": this sentence is unclear. How to "modulate the liquid flow" (i.e., how to change the flow velocity over time) when the PDMS adapter is already placed in the culture dish without changing the injection rate? Do you rather mean that the shape of the recess structure allows to select the velocity profile of the liquid flow?
- Page 3 Line 117, "at a constant speed": the simulation in the Figure 4C shows that the velocity profile provided by the U-shaped recess structure is not constant, but is weakly changing along the diameter. As a consequence, "at an almost constant speed" seems to be a more appropriate term.
- Figure 2E and 2F: the thickness of the top surface in blue should be reduced to suit with the thickness of the top surface in the Figures 2G and 3A.
- Figure 2G: important dimensions may be indicated in the figure, such as the 500 µm thickness and the 2 mm width of the recess structure, the 2 mm thickness of the PDMS bottom membrane, and the 5 mm width of the edge of the top surface.
- Page 6 Line 199, "Culture experiments were carried out at three times (n = 3)”: was the cell proliferation experiment performed three times, or was the same experiment observed three times?
In the former case, please remove "at": " Culture experiments were carried out three times (n = 3)”.
In the latter case, the evolution of the cell population should be shown in a graph (Occupied area (in absolute number) versus Time), instead of the histograms in the Figure 6B.
- Figure 6: the confluence (i.e., collective cell area) was measured in the areas i, ii and iii. In the case of the U-shaped recess structure, the simulation in the Figure 4A shows that the flow velocity is higher at those positions. As a consequence, is the cell confluence close to the inlet and close to the outlet also influenced at those positions?
Minor corrections:
- Page 2 Line 57: “in within microchannels". Please remove "in" or "within".
- Page 3 Line 98: how to understand “being attached centered under the 35-mm culture dish”? “being attached and centered using double-sided tape under the 35-mm culture dish"?
- Page 3 Line 109-110, “Similar holes were made on each counter electrode side to form an outlet": the term “counter electrode" is certainly an error, as there is no electrode in the perfusion system.
- Please introduce a space in “Figure1" ("Figure 1"), "Figure2" ("Figure 2"), “Figure3" ("Figure 3") in the figure legends (Lines 114, 125 and 145). The space is present in the legends of the other figures.
- Page 4 Lines 127-128 (legend of the Figure 2), "Place a 60-mm culture dish in the center of the 35-mm culture dish to which the circular arc was attached using double-sided tape": the correct sentence is likely "Place the 35-mm culture dish to which the circular arc was attached in the center of a 60-mm culture dish using double-sided tape”.
- Figure 3A, "Outlet/Disc harge plug": please correct the word "Discharge".
- Page 6 Line 189: "The cell was then imaged" --> “The cells were then imaged".
- Page 7 Line 223 (legend of the Figure 6): "and then the relative areas [with a "s"] at i and iii were [plural form] obtained".
- Page 7 Line 226 (legend of the Figure 6), "at pints i – iii": please correct the word "points".
Author Response
Response to Reviewer 2 Comments
December 27th, 2018
From: Reiko Yasuda
Graduate School of Agricultural Science, Kobe University
1-1 Rokkodai, Nada-ku, Kobe, 657-8501, Japan
Tel: +81-78-803-5928
r.yasuda@icomes.co.jp
Dear Sir,
We thank you for fruitful suggestions, especially for suggesting the better terms and sentences. Your valuable comments helped me very much to improve the quality as well as the readability of the paper. On the basis of your comments, I added new description. I hope the revision could meet the conditions to be considered as the publication. Please see attached word file for details of response to comment.

Reviewer 3 Report
The manuscript titled “Fabrication of a Novel Culture Dish Adapter with Recess Microstructure for Flow Control in a Closed Environment” describes a PDMS culture dish adapter with a recess microstructure for flow control compatible with commercially available culture dishes. The authors designed U-shaped and I-shaped adapters, and studied the effects of groove structure on medium flow as well as the effects of recess microstructures on cell proliferation.
The adapter was fabricated by replica molding using PDMS with recess microstructures 2 mm wide and 500 µm deep. The adapter allowed for oxygen permeability enabling cell culture as well as a thin transparent membrane (2 mm) in the center to allow for microscopy. The amount of medium in the culture dish can be adjusted by raising the adapter with a spacer.
The authors present a novel method for controlling the culture medium flow in a dish-based cell culture. I recommend the manuscript for publication following the suggested improvements below:
1) Please comment on how the CD adapter can be used without contamination. Comment on the measures you take to minimize the risk of contamination for the cell culture.
2) Figure 4C and 4D: The units on the y-axis are already in m/s. But the y-axis label reads “speed (x10^-7 m/s)”. Perhaps it can be changed as “Flow velocity (m/s)”. Alternatively the units of the y-axis can be changed.
3) Please label the x axis of Figure 6B as point i, ii, iii rather than “on holding jig”, “in center of groove” and “below holding jig”.
4) The experiments are performed at a single flow rate, and at this flow rate, the U-shaped recess microstructure leads a more uniform cell proliferation. Please comment on whether this would be true at lower or higher flow rates. For instance, the proliferation rate should invert (lower proliferation in the middle in comparison to the side regions) at high flow rates for the I-shaped structure due to the excessive shear stress applied to cells.
Minor:
1) Line 57: Please correct “experiments in within microchannels”.
2) Line 164: Please correct “when the flow velocity was 1.4 µL/min” as “when the flow rate was 1.4 µL/min”.
3) Line 164-165: Please correct: “Figure 4C and 4D show flow velocity distributions in the z-z' sections in Figure 3A and 3B.” as “Figure 4C and 4D show flow velocity distributions in the z-z' sections in Figure 4A and 4B.”
4) Line 189: Please correct: “The cell was” as “The cells were.
5) Line 191: There is a reference for “Figure 5B”, but Figure 5 is not labeled as A and B.
6) Line 199: Please correct “Culture experiments were carried out at three times” as “Culture experiments were carried out three times”.
Author Response
Response to Reviewer 3 Comments
December 27th, 2018
From: Reiko Yasuda
Graduate School of Agricultural Science, Kobe University
1-1 Rokkodai, Nada-ku, Kobe, 657-8501, Japan
Tel: +81-78-803-5928
r.yasuda@icomes.co.jp
Dear Sir,
We thank you for careful reading our manuscript and for giving accurate instructions. Your valuable comments helped me very much to improve the quality as well as the readability of the paper. On the basis of your comments, I added new description. I hope the revision could meet the conditions to be considered as the publication. Please see attached word file for details of response to comment.

Round 2
Reviewer 1 Report
The authors have carefully addressed all my comments and I am happy to recommend for publication in present form.